# Measuring the work environment among healthcare professionals: Validation of the Dutch version of the Culture of Care Barometer

**Susanne Maassen**[1,2]*, **Catharina van Oostveen**[3,4], **Anne Marie Weggelaar**[2], **Anne Marie Rafferty**[5], **Marieke Zegers**[6], **Hester Vermeulen**[7,8]

1 Department of Quality and Patientcare, Erasmus MC, University Medical Center, Rotterdam, The Netherlands, 2 TRANZO, Tilburg University, Tilburg, The Netherlands, 3 Erasmus School of Health Policy & Management, Erasmus University, Rotterdam, The Netherlands, 4 Spaarne Gasthuis Academy, Spaarne Gasthuis Hospital, Hoofddorp, Haarlem, The Netherlands, 5 Florence Nightingale Faculty of Nursing and Midwifery & Palliative Care, King's College London, London, United Kingdom, 6 Department of Intensive Care, Radboud Institute for Health Sciences, Radboud University Medical Center, Nijmegen, The Netherlands, 7 Radboud Institute for Health Sciences, Scientific Center for Quality of Healthcare (IQ healthcare), Radboud University Medical Center, Nijmegen, The Netherlands, 8 Faculty of Health and Social Studies, HAN University of Applied Sciences, Nijmegen, The Netherlands

* s.maassen@erasmusmc.nl

## Abstract

### Objectives

A positive work environment (WE) is paramount for healthcare employees to provide good quality care. To stimulate a positive work environment, employees' perceptions of the work environment need to be assessed. This study aimed to assess the reliability and validity of the Dutch version of the Culture of Care Barometer (CoCB-NL) survey in hospitals.

### Methods

This longitudinal validation study explored content validity, structural validity, internal consistency, hypothesis testing for construct validity, and responsiveness. The study was conducted at seven departments in two Dutch university hospitals. The departments were included based on their managers' motivation to better understand their employees' perception of their WE. All employees of participating departments were invited to complete the survey (n = 1,730).

### Results

The response rate was 63.2%. The content of the CoCB-NL was considered relevant and accessible by the respondents. Two factor models were found. First, confirmative factor analysis of the original four-factor structure showed an acceptable fit ($X^2$ 2006.49; *df* 399; p = <0.001; comparative fit index [CFI] 0.82; Tucker-Lewis index [TLI] 0.80; root mean square error of approximation [RMSEA] 0.09). Second, explanatory factor analysis revealed a five-factor model including 'organizational support', 'leadership', 'collegiality and teamwork',

**Data Availability Statement:** Data cannot be shared publicly because of Dutch privacy legislation. Data are available from the TRANZO

Tilburg University Institutional Data Access Committee for researchers who meet the criteria for access to confidential data. Request can be submitted via: https://dataverse.nl/dataset.xhtml?persistentId=doi:10.34894/WCX0WV.

**Funding:** The author(s) received no specific funding for this work.

**Competing interests:** The authors have declared that no competing interests exist.

'relationship with manager', and 'employee influence and development'. This model was confirmed and showed a better fit ($X^2$ 1552.93; *df* 395; p = < 0.00; CFI 0.87; TLI 0.86; RMSEA 0.07). Twelve out of eighteen hypotheses were confirmed. Responsiveness was assumed between the measurements.

## Conclusions

The CoCB-NL is a valid and reliable instrument for identifying areas needing improvement in the WE. Furthermore, the CoCB-NL appears to be responsive and therefore useful for longitudinal evaluations of healthcare employees' work environments.

## Introduction

A positive work environment (WE) is paramount for providing high-quality and safe patient care and for attracting and maintaining engaged healthcare professionals [1–7] The WE is the internal setting of the organization where employees work [8] and consists of the physical environment, culture, social climate, and job context [9]. A positive WE is characterized by respect, support, and trust between employees at all levels; effective collaboration and communication; recognition for good work; support from management; and a healthy work place [4, 9–11].

Healthcare organizations have become more aware of how the WE can influence patient and employee outcome measures and are committed to improving their WE [4, 12]. To better understand employees' experiences of their WE, surveys have been developed to gain a systematic insight into the WE. Periodic and valid measurements of the WE helps management learn from best practices and to understand which WE components need improving [4, 13–15]. However, the WE is multidimensional so is not easy to measure [9]. In addition, WEs differ within healthcare organizations and each department is likely to have its own characteristic WE [16, 17]. Achieving a positive WE is not just up to the members of one profession, but a challenge for the multidisciplinary team [6, 12]. This means that healthcare organizations need to use an instrument that encompasses important WE features from the perspective of all employees to measure the WE. This questionnaire should also be sensitive enough to differentiate between team and departmental responses and should be succinct and easy to understand by everyone [14].

A literature review [14] and Delphi study [9] found that the Culture of Care Barometer (CoCB) [15] is a complete, succinct, and applicable survey for all employees within healthcare organizations. The CoCB is designed to assess the WE within healthcare organizations. Rafferty et al. [15] described a positive WE as 'a caring culture'; a place where employees feel valued, respected, and supported and where relationships between employees, management, teams, and departments are good and where concerns can be discussed without fear of blame. The CoCB is a diagnostic and dialogic tool comprising 30 positively formulated items on four factors (organizational values, team support, relationships with colleagues, and job constraints) and one open question (what, if any, action needs to be taken to improve the culture of care environment where you work?) [15].

The validity and reliability of measurement instruments is determined by multiple measurement properties, each of which requires a different study design (Fig 1) [18, 19]. Rafferty et al. [15] determined content validity of the CoCB by asking healthcare professionals to assess the relevance and clarity of the items. Based on this, Rafferty et al. [15] defined the structural

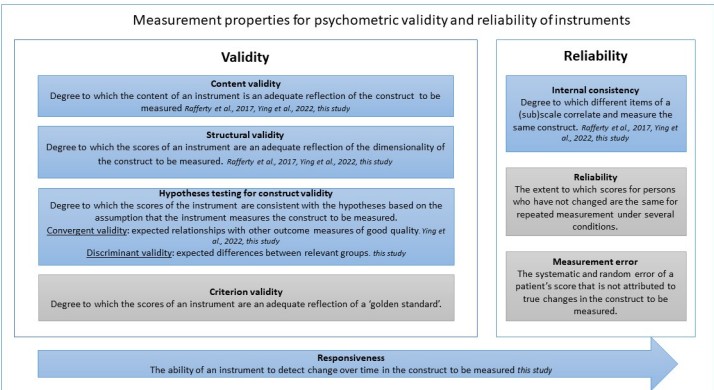

**Fig 1. Overview measurement properties based on COSMIN guideline.** In italics is indicated which studies have examined this measurement property.

validity and internal consistency of the CoCB using exploratory factor analysis (EFA). They found a four-factor structure with sufficient factor loadings > 0.40 [15, 18, 20]. These four factors were organizational values (12 items, 0.84–0.40), team support (11 items, 0.87–0.40), relationships with colleagues (4 items, 0.81–0.56), and job constraints (3 items, 0.79–0.41) [15]. The factors' internal consistency was proven by Cronbach's alphas between 0.70 and 0.93 [15].

The usability of a factor model found in an EFA is rarely absolute and finding a single model in a single data set does not prove that other plausible models do not also fit [20]. Therefore, EFA results need to be further evaluated by confirmatory factor analysis (CFA) on new data, to see whether other plausible models occur. Recently, Ying, Fitzpatrick [21] performed a CFA to validate the Chinese version of the CoCB. This CFA showed that the four-factor structure had acceptable fit (Chi-square [χ2] = 5975.22, 399 df, comparative fit index [CFI] 1.00, Tucker-Lewis index [TLI] 1.00, root mean square error of approximation [RMSEA] 0.07) and good internal consistency with Cronbach's alphas between 0.77 and 0.96 [21]. However, the authors did not examine alternative factor models. Additionally, Ying, Fitzpatrick [21] examined construct validity by testing hypotheses based on the relationships between the CoCB and job satisfaction and organizational culture. They found evidence of good content, structural and construct validity, and internal consistency between the English CoCB and the Chinese CoCB [15, 21]. However, the reliability and validity of the Dutch version (the CoCB-NL) has not yet been examined. Furthermore, no information is known about the responsiveness of the CoCB. The aim of the present study was to assess the reliability and validity of the CoCB-NL to determine whether broad uptake of the instrument should be recommended. For cross-cultural validation, the content and structural validity and internal consistency of the CoCB-NL were also examined. The instrument was further validated by assessing convergent validity, discriminant validity, and responsiveness.

## Methods

### Design

This validation study contained two parts and addressed five measurement properties:

1. Translation and cultural adaptation by determining the content validity of the translated CoCB items in the Dutch hospital context.

2. A longitudinal survey to confirm structural validity and internal consistency, testing hypotheses to determine the convergent validity and discriminant validity, and assessment of responsiveness of the CoCB-NL.

The study was developed and reported according to the Consensus-Based Standards for the Selection of Health Measurement Instruments (COSMIN) study design checklist [19] and reporting guideline [22].

## Setting and participants

The study was conducted in two Dutch hospitals that are part of the researchers' network and purposively approached hospitals that were planning to improve their work environment: the Radboud University Medical Center and the Erasmus University Medical Center. The survey was distributed to all employees working in the following departments at these hospitals: intensive care unit (ICU) (2), pediatric surgery (1), oncology (1), pharmacy (1), clinical chemistry (1), and endoscopy (1). The departments were included based on their managers' motivation to better understand their employees' perception of the WE.

## Part 1: Translation and content validity

The CoCB was translated into Dutch according to the translation procedure described by Sousa and Rojjanasrirat [23]. Three Dutch healthcare professionals (nurse, physiotherapist and researcher) independently provided the forward translation. These versions were compared by the research team (SM, AMW, CvO, HV) and checked for semantic and cross-cultural equivalence. Context-specific terminology such as 'trust' was changed to 'organization'. This first version of the instrument was back-translated into English by three independent native UK English speakers and was compared with the original UK version [15]. Discrepancies were discussed in the research team and resolved in cooperation with one of the co-authors and co-developer of the CoCB (AMR).

To determine content validity, 50 hospital employees, working in various positions in one of the hospitals affiliated with the researchers, were asked to assess the relevance and accessibility of the CoCB-NL items using a dichotomous yes/no scale. The threshold for content validity at the item level was set at an 80% 'yes' score [23]. Thirty-seven respondents (response rate 74%) (S1 Table) completed the content validity questionnaire and the threshold was reached for 26/30 items with scores between 89.2% and 100% on relevance and comprehensibility (Table 1). The four items that did not reach the threshold were: 'Unacceptable behavior is consistently tackled' (78.4%), 'The organization values the service we provide' (76.3%), 'I have the resources I need to do a good job' (71.1%), and 'Managers at the top of the organization know how things really are'(62.2%). Considering the expected relevance of these items for practice, we decided to further evaluate them in the second part of the study. Footnotes were added to the questionnaire to clarify what was meant by 'the organization' and who was meant by 'managers at the top'.

## Part II: Survey

The survey contained the CoCB-NL to which the nine-item version of the Utrecht Work Engagement scale (UWES-9) [24] and the team and safety climate subscales of the Safety Attitudes Questionnaire (SAQ-NL) [25, 26] were also added for hypothesis testing [18]. The UWES-9 and SAQ-NL are psychometrically sound instruments [24–26]. Both the CoCB-NL and SAQ-NL use a five-point Likert scale ranging from 'do not agree at all' to 'totally agree' [15, 25, 26]. The UWES-9 is rated on a seven-point scale ranging from 'never' to 'always' [24].

**Table 1. Content validity assessment on comprehensibility.**

| No | N = 37 | % YES score |
|---|---|---|
| 1 | I have the resources i need to do a good job | 71.10% |
| 2 | I feel respected by my coworkers | 97.30% |
| 3 | I have sufficient time to do my job well | 100% |
| 4 | I am proud to work in this Trust | 97.30% |
| 5 | My line manager treats me with respect | 94.60% |
| 6 | The Trust values the service we provide | 76.30% |
| 7 | I would recommend this Trust as a good place to work | 92.10% |
| 8 | I feel well supported by my line manager | 97.30% |
| 9 | I am able to influence the way things are done in my team | 97.30% |
| 10 | I feel part of a well managed team | 97.30% |
| 11 | I know who my line manager is | 97.30% |
| 12 | Unacceptable behaviour is consistently tackled | 78.40% |
| 13 | There is strong leadership at the highest level in the Trust | 89.20% |
| 14 | When things get difficult, I can rely on my colleagues | 94.60% |
| 15 | Trust managers know how things really are | 62.20% |
| 16 | I feel able to ask for help when I need it | 97.30% |
| 17 | I know exactly what is expected of me in my job | 94.60% |
| 18 | I feel supported to develop my potential | 100% |
| 19 | A positive culture is visible where I work | 97.30% |
| 20 | The people I work with are friendly | 97.30% |
| 21 | My line manager gives me constructive feedback | 100% |
| 22 | Staff successes are celebrated by the Trust | 91.90% |
| 23 | The Trust listens to staff views | 91.90% |
| 24 | I get the training and development I need | 97.30% |
| 25 | I am able to influence how things are done in the Trust | 91.90% |
| 26 | The Trust has a positive culture | 89.20% |
| 27 | I am kept well informed about what is going on in our team | 100% |
| 28 | I have positive role models where I work | 89.20% |
| 29 | I feel well informed about what is happening in the Trust | 97.30% |
| 30 | My concerns are taken seriously by my line manager | 97.30% |

Being latent variables, the scores are calculated as subscales in the CoCB-NL and SAQ-NL and as scales in the UWES-9. The survey was sent out twice, at an interval of 9 to 12 months, in line with the continuous improvement cycle of the respective department. Data from the first survey were used to assess the structural validity, internal consistency, convergent validity, and discriminant validity. Data from the second survey were used to assess responsiveness.

To minimize the measurement burden, administration of the UWES-9 and SAQ-NL subscales was optional. All departments chose to extend the survey with the SAQ safety climate subscale. One ICU also added the team climate subscale. The pharmacy, clinical chemistry, pediatric surgery, and oncology departments volunteered to measure engagement with the UWES-9.

All employees were informed about the survey by their manager before receiving an invitation to participate by email from the researchers. Data were collected for six weeks via the online survey tool LimeSurvey®. Automatic reminders were sent after two and four weeks to those participants who had not yet completed the survey. All questions on the CoCB-NL, UWES-9, and SAQ-NL subscales were obligatory. Questions about personal characteristics

like age group, gender, type of contract, mother language, and profession were voluntary as required by the Dutch privacy legislation. Data collection took place in 2018 and 2019.

**Ethical considerations.** Ethical approval was waived by the Medical Ethic Committee Brabant (NW2023-32) since this study did not include patients and did not affect the participants' wellbeing. The data collection and data storage plan was approved by the local GDPR committee of the Erasmus Medical Center Rotterdam. Participation to the study was anonymous and voluntary. Participants were informed about this both in the announcement e-mail and at the beginning of the questionnaire itself (page 1). By proceeding to the second page with the substantive questions, participants gave consent. Technically, a strict separation was made between the LimeSurvey® input file containing personal data and the output file, which did not include this data. Researchers only had access to the output file.

**Structural validity and internal consistency.** Structural validity and internal consistency were assessed in the CoCB-NL in two stages. First, an EFA was performed to see if other plausible factor models exist. Second, both the UK factor model and other potential models were evaluated with a CFA. Internal consistency was calculated in all models.

**Convergent validity, discriminant validity, and responsiveness.** In total, 18 hypotheses were formulated for testing. Fifteen hypotheses on both the item level (1–9) and subscale level (10–15) were formulated to assess convergent validity (Table 2). These hypotheses were based on associations between a poor perceived WE and low employee engagement [3, 17] or poor safety climate [4, 7, 27] and an expected association between the WE and team climate [27]. Hypotheses to test discriminant validity were based on existing differences in positions and secondary working conditions between professional groups [18]. For example, the influence of nurses on hospital policy was perceived as less than that of physicians [28]. Three hypotheses (16–18) were formulated based on the differences in position and secondary working conditions between physicians, nurses, pharmacists, and laboratory staff.

The responsiveness of the CoCB-NL for detecting change over time was explored by comparing the development and trends in the CoCB-NL, UWES-9, and SAQ-NL safety climate and team climate between the first and second measurement on the department level.

**Analysis.** Data analysis was performed using IBM SPSS Statistics version 25 and IBM SPSS AMOS version 25. For optimal use of SPSS AMOS, only complete cases were analyzed. The minimum sample size was set at seven times the number of items [19, 29]. Data were analyzed on the total sample and department level. For each CoCB-NL item normal distribution as a prerequisite for further analysis was calculated (S2 Table). EFA was performed without a predefined number of factors using principal component analysis with a varimax rotation method, including Kaiser normalization. Factor loadings above 0.4 were considered acceptable [30, 31]. The CFA model fit was assessed by multiple fit indices including $X^2$, degrees of freedom (df), CFI, TLI, RMSEA, and the upper and lower 90% confidence interval (CI) of the RMSEA. These indices comprehensively evaluate model fit, with CFI and TLI values $>0.95$ and RMSEA values $<0.06$ indicating excellent model fit [32–35]. For an instrument with 30 items, the df should be 405 minus the number of factors $m$ ($m \times (m - 1)/2$) and the ratio of $X^2$ to df should be $\leq 2$ [32, 33, 35]. No post hoc modifications were applied in the CFA [33, 35]. Internal consistency of the factors in both models was assessed with Cronbach's α coefficient and good internal consistency was indicated by α values between 0.70 and 0.95 [18, 34].

The hypotheses for convergent validity of CoCB-NL items were tested using Spearman's correlation coefficients for ordinal data. The hypotheses on CoCB-NL subscales were tested using Pearson correlation coefficients for continuous data. Hypotheses were accepted or rejected based on cut-off values defined in the COSMIN guidelines for construct validity assessment [18, 34]. All but one hypothesis addressed related but dissimilar constructs; therefore, correlation coefficients of 0.30–0.50 between the CoCB-NL and UWES-9 or SAQ-NL

**Table 2. Hypothesis testing for construct validity.**

| Convergent validity: hypothesis on item level | N | Spearman's rho ρ | P-value (2-tailed) |
|---|---|---|---|
| H1. Moderate correlation between the items 'I feel well supported by my line manager (CoCB-NL item 8)' and 'I have the support I need from other personnel to care for patients (SAQ-NL TC item 4)'. | 297 | 0.11 | 0.051 |
| H2 Moderate correlation between the items 'I am able to influence the way things are done in my team (CoCB-NL item 9)'and 'Nurse input is well received in this clinical area (SAQ-NL TC item 1)'. | 297 | 0.36 | <0.000 |
| H3 Moderate correlation between the items 'The organization listens to staff views (CoCB-NL item 23)'and 'Nurse input is well received in this clinical area (SAQ-NL TC item 1)'. | 297 | 0.41 | <0.000 |
| H4 Moderate correlation between the items 'I am able to influence how things are done in the organization (CoCB-NL item 25)' and 'Nurse input is well received in this clinical area (SAQ-NL TC item 1)'. | 297 | 0.34 | <0.000 |
| H5 Moderate correlation between the items 'I feel part of a well-managed team (CoCB-NL item 10)' and 'The physicians and nurses here work together as a well-coordinated team (SAQ-NL TC item 6)'. | 297 | 0.23 | <0.000 |
| H6 Moderate correlation between the items 'I feel able to ask for help when I need it (CoCB-NL item 16)' and 'It is easy for personnel here to ask questions when there is something that they do not understand (SAQ-NL TC item 5)'. | 297 | 0,30 | <0.000 |
| H7 Strong correlation between the items 'My line manager gives me constructive feedback (CoCB-NL item 21)' and 'I receive appropriate feedback about my performance (SAQ-NLSC item 10)'. | 971 | 0.57 | <0.000 |
| H8 Moderate correlation between the items 'My concerns are taken seriously by my line manager (CoCB-NL item 30)' and 'I am encouraged by my colleagues to report any patient safety concerns I may have (SAQ-NL SC item 12)'. | 971 | 0.25 | <0.000 |
| H9 Moderate correlation between the items 'I am proud to work in this organization (CoCB-NL item 4)' and 'I would feel safe being treated here as a patient (SAQ-NL SC item 7)'. | 972 | 0.37 | <0.000 |
| **Convergent validity: hypothesis on subscale level** | **N** | **Pearson correlation *r*** | ***P*-value (2-tailed)** |
| H10 There is a moderate correlation between the CoCB-NL subscale 'collegiality and teamwork' and the SAQ-NL subscale 'team climate'(SAQ-NL TC total score). | 297 | 0.54 | <0.000 |
| H11 There is a moderate correlation between the CoCB-NL subscale 'relationship with manager' and the SAQ-NL subscale 'safety climate' (SAQ-NL SC total score). | 972 | 0.53 | <0.000 |
| H12 There is a moderate correlation between the CoCB-NL subscale 'leadership' and the SAQ-NL subscale 'safety climate' (SAQ-NL SC total score). | 972 | 0.64 | <0.000 |
| H13 There is a moderate correlation between the CoCB-NL subscale 'collegiality and teamwork' and the UWES-9 total score. | 455 | 0.46 | <0.000 |
| H14 There is a moderate correlation between the CoCB-NL subscale 'relationship with manager' and the UWES-9 total score. | 455 | 0.41 | <0.000 |
| H15 There is a moderate correlation between the CoCB-NL subscale 'employee influence and development' and the UWES-9 total score. | 455 | 0.46 | <0.000 |
| **Discriminant validity: hypothesis on subscale level** | | **N** | **Mean score (std. deviation)** | ***P*-value (2-tailed)** |
| H16 Physicians will have a higher score on the CoCB-NL subscale 'employee influence and development' than nurses. | Physicians / Nurses | 182 / 471 | 3.45 (0.65) / 3.24 (0,59) | <0.000 |
| H17 Nurses will have a higher score on the CoCB-NL subscale 'employee influence and development' than pharmacists. | Nurses / Pharmacists | 471 / 67 | 3.24 (0.59) / 2.99 (0.76) | 0.002 |
| H18 Nurses will have a higher score on the CoCB-NL subscale 'employee influence and development' than laboratory staff. | Nurses / Laboratory staff | 471 / 98 | 3.24 (0.59) / 2.81 (0.70) | <0.000 |

Correlation <0.30 = weak; 0.30–0.50 = moderate; >0.5 = strong

were considered acceptable [18, 34]. Only hypothesis 7 compares two nearly equal items, so we expected a strong correlation (>0.50). The hypotheses for discriminant validity were tested using a Student t-test for independent samples with a 95% CI. A p-value of <0.05 indicated a significant difference between groups. The CoCB-NL's ability to detect change over time at the department level was analyzed using a Student t-test for independent samples with a 95% CI. The trend of each outcome measure was visually analyzed as a bar chart.

## Results

All employees of the participating departments (n = 1,730) were invited to participate in the study. Of these, 1,094 (63.2%) respondents completed survey 1 and 1,062 of these were complete cases (Table 3). Most respondents were female (71.9%) and were equally distributed among age groups and types of contracts. Eight professional groups participated in the survey; nurses were the largest group (47.9%), followed by physicians (18.5%). Survey 2 was completed by 590 respondents (48.4%).

### Structural validity and internal consistency

The CFA of the original CoCB-based factor model had factor correlations between 0.63 and 0.84. The standardized item loadings of the factors ranged from 0.43 to 0.85 (Table 4). The model fit indices did not reach the thresholds for an excellent fit ($X^2$ 2006.49; $df$ 399; p = <0.00; CFI 0.82; TLI 0.80; RMSEA 0.09; 90% CI 0.08–0.09). Three out of four factors showed internal consistency with Cronbach's α coefficients of 0.76–0.91. For the factor 'job constraint' a Cronbach's α coefficient of 0.51 was found.

The EFA identified a five-factor model (Table 5). The first factor contained six items that emerged from the original factors 'organizational values' and 'job constraint' (items 1, 3, 4, 6, 7, and 26; loadings 0.47–0.67). The items reflected the respondents' experience of organizational support and their valuation of the organization so we named this factor 'organizational support'. The second factor included seven items concerning 'leadership' (items 10, 12, 13, 15, and 27–29; loadings 0.46–0.66). These items were a combination of items from three factors in the original model. This was also the case for the third factor, which included six items related to 'collegiality and teamwork' (2, 14, 16, 17, 19, and 20; loadings 0.54–0.72). The fourth factor included five items about the 'relationship with manager' and originated from the original factor 'team support' (5, 8, 11, 21, and 30; loadings 0.63–0.77). The last factor contained six items on 'employee influence and development (9, 18, 22–25; loadings 0.47–0.77). These items emerged from the factors 'organizational values' and 'team support' in the original model. The correlations between the five factors ranged from 0.68 to 0.88. The model fit indices were $X^2$ 1552.93, $df$ 395, p = < 0.00, CFI 0.87, TLI 0.86, and RMSEA 0.07 (90% CI 0.07–0.08). Although the CFI, TLI, and RSMEA did not reach the threshold for an excellent model fit, the values were more favorable than those of the four-factor model. All factors showed internal consistency with Cronbach's α coefficients varying between 0.79 and 0.88.

### Convergent validity, discriminant validity, and responsiveness

For convergent validity, six out of nine hypotheses were accepted on the item level (2, 3, 4, 6, 7, and 9) based on strong correlation coefficients (Table 2). The hypotheses 2, 3, 4, 6, and 9 showed moderate coefficients between $r$ 0.30 (6) and $r$ 0.41 (3). Hypothesis 7 revealed the expected strong correlation ($r$ 0.57). The remaining three hypotheses showed weak correlation coefficients on the item level (hypotheses 1, 5 and 8; $r$ 0.11 –$r$ 0.25) and were therefore rejected. Three out of six factor-level hypotheses were accepted (13, 14, and 15 out of 10–15). The hypotheses concerning work engagement (UWES-9) and CoCB-NL (13–15) had moderate correlation coefficients of $r$ 0.41 to $r$ 0.46 and were thus accepted. A strong correlation coefficient was observed between safety climate and the CoCB-NL factors 'relationship with manager' and 'leadership' ($r$ 0.53 and $r$ 0.64) as well as between team climate and the CoCB-NL factor 'collegiality and teamwork' ($r$ 0.54) instead of the expected moderate correlation, so these hypotheses were rejected.

All three hypotheses for discriminant validity (16–18) were accepted based on a significant Student t-test (p <0.01). The mean scores for the factor 'employee influence and development'

**Table 3. Sample characteristics.**

| | T0 | T0 | T1 | Comparison for departments participating on both T0 and T1. |
|---|---|---|---|---|
| | total sample | sample for responsiveness* | | |
| | % | % | % | P value** |
| **Sample size** | N = 1,062 | N = 792 | N = 590 | |
| **Department** | | | | 0.124 |
| Clinical chemistry | 12.3 | 16.2 | 12.2 | |
| Endoscopy | 6.1 | NA | NA | |
| Intensive care A | 34.3 | 45.1 | 47.8 | |
| Intensive care B | 17.0 | 22.3 | 21.0 | |
| Oncology | 12.7 | NA | NA | |
| Pediatric surgery | 5.1 | NA | NA | |
| Pharmacy | 12.5 | 16.4 | 19.0 | |
| **Gender** | | | | <0.001 |
| Male | 25.7 | 27.7 | 26.5 | |
| Female | 71.9 | 69.8 | 71.8 | |
| Neutral | 2.3 | 2.5 | 1.6 | |
| **Age group** | | | | 0.902 |
| 16–29 | 19.0 | 16.5 | 16.5 | |
| 30–39 | 26.0 | 25.9 | 24.4 | |
| 40–49 | 20.6 | 20.7 | 22.3 | |
| 50–59 | 24.6 | 26.9 | 25.9 | |
| 60–66 | 9.9 | 10.1 | 11.0 | |
| **Dutch as first language** | | | | 0.244 |
| Yes | 94.1 | 94.2 | 92.7 | |
| No | 5.9 | 5.8 | 7.3 | |
| **Contract** | | | | <0.001 |
| Full time | 48.0 | 49.8 | 47.7 | |
| Part time | 52.0 | 50.2 | 52.3 | |
| **Profession** | | | | 0.006 |
| Administrative staff | 5.8 | 5.8 | 6.5 | |
| Care assistants | 6.4 | 4.3 | 9.1 | |
| Laboratory technicians | 10.0 | 13.3 | 11.3 | |
| Managers | 2.8 | 2.7 | 3.0 | |
| Nurses | 47.9 | 46.5 | 45.3 | |
| Pharmacists | 6.8 | 9.1 | 10.7 | |
| Physicians | 18.5 | 15.9 | 10.4 | |
| Researchers | 1.8 | 2.4 | 3.7 | |

* Sample including only the departments that participated in the assessment of responsiveness.

** Tested with 95% confidence interval

ranged from 3.45 (physicians) to 2.81 (laboratory staff), and differed between professional groups (p <0.01).

Four departments completed two measurements during this study; n = 792 participants responded to survey 1 and n = 569 participants responded to survey 2. Both groups were comparable with regard to department, age group, and having Dutch as the first language (Table 3). The scores ranged from 2.93 (employee influence and development) to 4.14

**Table 4. Confirmative factor analysis and internal consistency of the four-factor model.**

| Label | Name n = 550 | Standardized loadings CFA | Cronbach's α | Correlation-coefficient ρ |
|-------|------|------|------|------|
| F1 | *Organizational values* | | 0.91 | F2: 0.84 F3: 0.63 F4: 0.77 |
| Q4 | I am proud to work in this organization | 0.62 | | |
| Q6 | The organization values the service we provide | 0.68 | | |
| Q7 | I would recommend this organization as a good place to work | 0.75 | | |
| Q13 | There is strong leadership at the highest level in the organization | 0.69 | | |
| Q15 | Managers in the top of the organization know how things really are | 0.60 | | |
| Q19 | A positive culture is visible where I work | 0.67 | | |
| Q22 | Staff successes are celebrated by the organization | 0.65 | | |
| Q23 | The organization listens to staff views | 0.74 | | |
| Q24 | I get the training and development I need | 0.55 | | |
| Q25 | I am able to influence how things are done in the organization | 0.72 | | |
| Q26 | This organization has a positive culture | 0.75 | | |
| Q29 | I feel well informed about what is happening in the organization | 0.71 | | |
| F2 | *Team support* | | 0.91 | F1: 0.84 F3: 0.73 F4: 0.76 |
| Q5 | My line manager treats me with respect | 0.80 | | |
| Q8 | I feel well supported by my line manager | 0.85 | | |
| Q9 | I am able to influence the way things are done in my team | 0.66 | | |
| Q10 | I feel part of a well-managed team | 0.67 | | |
| Q11 | I know who my line manager is | 0.43 | | |
| Q12 | Unacceptable behavior is consistently tackled | 0.56 | | |
| Q16 | I feel able to ask for help when I need it | 0.61 | | |
| Q18 | I feel supported to develop my potential | 0.68 | | |
| Q21 | My line manager gives me constructive feedback | 0.79 | | |
| Q27 | I am kept well informed about what is going on in our team | 0.65 | | |
| Q30 | My concerns are taken seriously by my line manager | 0.82 | | |
| F3 | *Relationship with colleagues* | | 0.76 | F1: 0.63 F2: 0.73 F4: 0.77 |
| Q2 | I feel respected by my coworkers | 0.67 | | |
| Q14 | When things get difficult, I can rely on my colleagues | 0.73 | | |
| Q20 | The people I work with are friendly | 0.65 | | |
| Q28 | I have positive role models where I work | 0.63 | | |
| F4 | *Job constraint* | | 0.51 | F1: 0.77 F2: 0.76 F3: 0.77 |
| Q1 | I have the resources I need to do a good job | 0.50 | | |
| Q3 | I have sufficient time to do my job well | 0.44 | | |
| Q17 | I know exactly what is expected of me in my job | 0.56 | | |

(collegiality and teamwork) for survey 1 and 3.00 (leadership) to 4.17 (employee influence and development) for survey 2.

Responsiveness was assumed because the bar charts presenting outcome measures between surveys 1 and 2 showed similar patterns in all departments (Fig 2). There was a significant

**Table 5. Exploratory and confirmative factor analysis and internal consistency of the five-factor model.**

| Label | Name | Factor loadings EFA n = 512 | Standardized loadings CFA n = 550 | Cronbach's α | Correlation-coefficient ρ |
|---|---|---|---|---|---|
| F1 | *Organizational support* | | | 0.79 | F2: 0.88<br>F3: 0.68<br>F4: 0.68<br>F5: 0.85 |
| Q1 | I have the resources I need to do a good job | 0.61 | 0.45 | | |
| Q3 | I have sufficient time to do my job well | 0.67 | 0.38 | | |
| Q4 | I am proud to work in this organization | 0.65 | 0.68 | | |
| Q6 | The organization values the service we provide | 0.56 | 0.70 | | |
| Q7 | I would recommend this organization as a good place to work | 0.65 | 0.81 | | |
| Q26 | This organization has a positive culture | 0.47 | 0.74 | | |
| F2 | *Leadership* | | | 0.84 | F1: 0.88<br>F3: 0.72<br>F4: 0.80<br>F5: 0.88 |
| Q10 | I feel part of a well-managed team | 0.52 | 0.70 | | |
| Q12 | Unacceptable behavior is consistently tackled | 0.63 | 0.60 | | |
| Q13 | There is strong leadership at the highest level in the organization | 0.54 | 0.69 | | |
| Q15 | Managers in the top of the organization know how things really are | 0.46 | 0.58 | | |
| Q27 | I am kept well informed about what is going on in our team | 0.65 | 0.69 | | |
| Q28 | I have positive role models where I work | 0.48 | 0.63 | | |
| Q29 | I feel well informed about what is happening in the organization | 0.66 | 0.72 | | |
| F3 | *Collegiality and teamwork* | | | 0.83 | F1: 0.68<br>F2: 0.77<br>F4: 0.69<br>F5: 0.69 |
| Q2 | I feel respected by my coworkers | 0.69 | 0.64 | | |
| Q14 | When things get difficult, I can rely on my colleagues | 0.71 | 0.69 | | |
| Q16 | I feel able to ask for help when I need it | 0.55 | 0.72 | | |
| Q17 | I know exactly what is expected of me in my job | 0.54 | 0.56 | | |
| Q19 | A positive culture is visible where I work | 0.60 | 0.76 | | |
| Q20 | The people I work with are friendly | 0.72 | 0.64 | | |
| F4 | *Relationship with manager* | | | 0.88 | F1: 0.68<br>F2: 0.80<br>F3: 0.69<br>F5: 0.76 |
| Q5 | My line manager treats me with respect | 0.76 | 0.84 | | |
| Q8 | I feel well supported by my line manager | 0.77 | 0.87 | | |
| Q11 | I know who my line manager is | 0.63 | 0.42 | | |
| Q21 | My line manager gives me constructive feedback | 0.72 | 0.82 | | |
| Q30 | My concerns are taken seriously by my line manager | 0.67 | 0.84 | | |
| F5 | *Employee influence and development* | | | 0.85 | F1: 0.85<br>F2: 0.88<br>F3: 0.69<br>F4: 0.76 |
| Q9 | I am able to influence the way things are done in my team | 0.52 | 0.68 | | |
| Q18 | I feel supported to develop my potential | 0.64 | 0.73 | | |
| Q22 | Staff successes are celebrated by the organization | 0.47 | 0.64 | | |

*(Continued)*

**Table 5.** (Continued)

| Label | Name | Factor loadings EFA n = 512 | Standardized loadings CFA n = 550 | Cronbach's α | Correlation-coefficient ρ |
|---|---|---|---|---|---|
| Q23 | The organization listens to staff views | 0.63 | 0.76 | | |
| Q24 | I get the training and development I need | 0.65 | 0.61 | | |
| Q25 | I am able to influence how things are done in the organization | 0.77 | 0.76 | | |

change in 'organizational support' (department A p = 0.042 and department C p = 0.026) and 'safety climate' (department D p = 0.045) between survey 1 and 2.

## Discussion

This study aimed to assess the cross-cultural validity of the CoCB in Dutch hospitals and to psychometrically validate the instrument. The content of the CoCB-NL was considered relevant and accessible by the respondents. The original four-factor model, including 'organizational values', 'team support', 'relationships with colleagues', and 'job constraints', and the new five-factor model, including 'organizational support', 'leadership', 'collegiality and teamwork', 'relationship with manager', and 'employee influence and development', were found to be sufficiently valid and reliable for use in Dutch hospitals. All hypotheses for discriminant validity and more than half of the hypotheses for convergent validity were confirmed, which provided satisfactory construct validity. Furthermore, the CoCB-NL appears to be responsive in detecting changes over time so is a useful instrument for longitudinal evaluations of the WE.

### Reflections on structural validity and internal consistency

CFA of the four-factor model was performed in accordance with the original factor structure reported by Rafferty et al. [15]. This model showed an acceptable fit in our Dutch sample; however, this fit was not as strong as that found for the Chinese version of the CoCB [21]. We performed an EFA to check for other plausible models as recommended by several authors [20, 32, 33, 35] and found a five-factor model. The CFA of this model showed a better fit in our

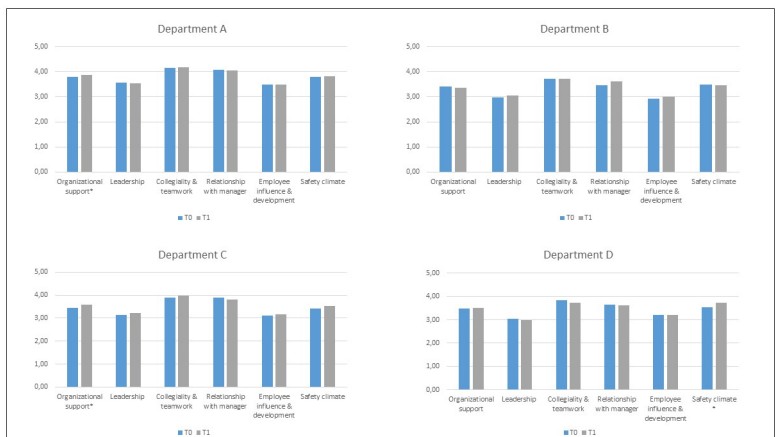

**Fig 2. Bar charts on department level with difference between survey 1 (T0) and 2 (T1).** * = a significant change between T0 and T1 with p < 0.05 (95% CI).

Dutch sample. These differences between the four- and five-factor structures can be explained by the context in which the validation studies were conducted and the level at which the data were analyzed. The original four-factor instrument was developed in the UK and measures the WE at the trust, management, and ward level [15]. In the UK, hospital trusts can comprise multiple hospitals, such as the five hospitals making up Barts Health in London [36]. In contrast, hospitals in the Netherlands are independent entities. Therefore, we translated the original instrument into one that can measure the WE in a single organization, as was also done in the Chinese translation [21]. This resulted in factors on organizational foundations like 'leadership' and 'employee influence and development' in the Dutch version. These factors have been defined as essential elements of the WE [9].

Based on the psychometrics, the five-factor model is preferable in the Dutch context whereas the four-factor structure is better for international benchmarking. This justifies our decision not to drop or adjust any of the original CoCB items, although item 11 ('I know who my line manager is') does qualify for adjustment. The weak presentation of this item in both models can be explained by the lack of variation in the responses (95.8% agreement). If future research points out that this item remains weak in other language versions of the CoCB, dropping this item could be considered.

### Reflections on convergent validity, discriminant validity, and responsiveness

In total, 67% (12 out of 18) of the tested hypotheses for construct validity were confirmed. Three hypotheses were rejected because contrary to the expected moderate correlations, strong correlations were found between factors of the CoCB-NL and the SAQ-NL. This is commensurate findings of Brubakk et al. [37] who found that positive work environments associate with a positive safety climate and favorable patient safety outcomes. The remaining three hypotheses were rejected because 1) support from managers did not appear to be a prerequisite for the support of colleagues or other personnel, and 2) working as a well-coordinated team did not necessarily mean being part of a well-managed team.

Overall, the CoCB-NL had moderate to high scores. Participants scored the organizational support factor evenly; however, there was variation in department specific factors. This is important for two reasons. First, because participants generally agreed with the positively formulated items used to score their WE. Second, because the variation in scores demonstrated the power to detect best and bad practices, which created learning opportunities among departments [16, 25].

### Implications for practice and future research

These findings provide first insights into the WE. Now, further investigation on how to improve the WE is needed. Huebner and Zacher [38] found that active follow-up discussions and action planning after employee surveys benefitted improvement of the WE. The open question in the CoCB-NL could be a useful place to start such active follow-up discussions [39]. This should be part of continuous efforts to improve the WE since a positive WE is a cornerstone for high-performance hospitals [4, 10]. Our finding that the CoCB-NL is responsive indicates that it can also be used for long-term evaluation of WE improvement.

### Limitations

This study has some limitations. First, although substantial effort was made to validate the instrument by assessing five measurement properties, the instrument's measurement error is still unknown (Fig 1). Establishing measurement error requires two measurements in the same

group under stable conditions [19]. These conditions are difficult to achieve because employees' perceptions of the WE is influenced by multiple factors, such as staffing ratios [1] or a pandemic [40]. Second, responsiveness was determined on the team level. Although determining responsiveness on the individual employee level may have given stronger evidence [41], the current results show promising responsiveness. Future studies may continue to measure responsiveness at the team level to ensure the privacy and safety of respondents, which is a requirement of these surveys. Third, there was an interval of 9–12 months between survey 1 and survey 2, which might not have been long enough to significantly change the participants' experiences of their WE. However, we did observe a significant difference in one factor among the three departments. Fourth, data were collected in two large-scale university hospitals and therefore may not be generalizable to other type of institutions. These large organizations are characterized by an extensive management line, politics, and hierarchy. This means our respondents could distinguish between the 'top of the organization' and the 'line manager', as requested in several items. This may be more difficult for respondents from smaller organizations.

## Conclusions & recommendations

This study demonstrated that the CoCB-NL is sufficiently valid and reliable for assessing employees' perceptions of their WE. Both a four-factor and five-factor model showed acceptable fit. We recommend using the five-factor model in the Dutch setting based on our findings and the four-factor model in the international setting based on previous validations in two other languages.

A positive WE is paramount for good patient outcomes and for attracting and keeping healthcare staff. Therefore, more research is needed to continuously improve the WE. Continuous improvement of the WE should start with discussions based on the results and open question in the CoCB-NL.

## Supporting information

**S1 Table. Sample characteristics of content validity on comprehensibility.**
(PDF)

**S2 Table. CoCB-NL item normal distribution.**
(XLSX)

**S3 Table. Dutch language version of the CoCB-NL.**
(XLSX)

## Acknowledgments

The authors thank Hilco van Elten, Erasmus School of Health Policy and Management, Rotterdam, The Netherlands, for his assistance with the confirmative factor analysis in SPSS AMOS. We also thank all participating teams and their management for their willingness and enthusiasm to participate.

## Author Contributions

**Conceptualization:** Catharina van Oostveen, Marieke Zegers, Hester Vermeulen.

**Formal analysis:** Susanne Maassen, Catharina van Oostveen.

**Investigation:** Susanne Maassen, Marieke Zegers.

**Methodology:** Susanne Maassen, Catharina van Oostveen, Marieke Zegers, Hester Vermeulen.

**Project administration:** Susanne Maassen.

**Supervision:** Catharina van Oostveen, Anne Marie Weggelaar, Anne Marie Rafferty, Hester Vermeulen.

**Validation:** Susanne Maassen, Hester Vermeulen.

**Writing – original draft:** Susanne Maassen, Catharina van Oostveen.

**Writing – review & editing:** Catharina van Oostveen, Anne Marie Weggelaar, Marieke Zegers.

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
