## [Decision Letter · Decision Letter 0]

16 Aug 2023

PONE-D-23-07783"Measuring the work environment among healthcare professionals: validation of the Dutch version of the Culture of Care Barometer"PLOS ONE

Dear Dr. Susanne M. Maassen, MSc;

Thank you for submitting your manuscript to PLOS ONE. After careful consideration, we feel that it has merit but does not fully meet PLOS ONE’s publication criteria as it currently stands. Therefore, we invite you to submit a revised version of the manuscript that addresses the points raised during the review process.

We look forward to receiving your revised manuscript.

Kind regards,

Elif Ulutaş Deniz

Academic Editor

PLOS ONE

Journal Requirements:

Additional Editor Comments:

Manuscript ID PONE-D-23-07783 entitled "Measuring the work environment among healthcare professionals: validation of the Dutch version of the Culture of Care Barometer" which you submitted to the PLOS ONE, has been reviewed. The comments of the reviewer(s) are included at the bottom of this letter.

The reviewer(s) have recommended publication, but also suggest some revisions to your manuscript. Therefore, I invite you to respond to the reviewer(s)' comments and revise your manuscript.

Reviewer 1:

Recognising the centrality of a work environment that cares for and values its workforce for recruiting and retaining the workforce, achieving a high-performing organisation, and for safe and high-quality patient care, the aim of this study was to test the psychometric properties of a Dutch version of the Culture of Care Barometer.

The abstract was presented using headers and core details were included. There was no reference to a methodological framework for conducting this study.

The Introduction was succinct in contextualising this study- what characterises a positive work environment and why a positive work environment has positive outcomes for staff, patients, and wok organisations.

As a foundation for this study the authors conducted a systematic review on the psychometric evaluation of instruments measuring the work environment of healthcare professionals in hospitals, this identified the Culture of Care Barometer (Rafferty et al 2017) as an instrument that was worthy of further investigation to determine its validity and reliability for use in the Netherlands. Following on from this was a Delphi study by the authors that culminated in confirmation of 36 items by the experts.

For cross-cultural validation, the content and structural validity and internal consistency of the CoCB-NL were examined. The instrument was further validated by assessing convergent validity, discriminant validity, and responsiveness.

The design involved translation and cultural adaptation and a longitudinal survey to confirm structural validity and internal consistent, testing hypotheses to determine the convergent and discriminant validity and assessment of responsiveness.

The study was developed and reported according to the Consensus-Based Standards for the Selection of Health Measurement Instruments (COSMIN) study design checklist [19] and reporting guideline [22].

2 Dutch hospitals participated- the sampling approach was not reported- and a convenience sample of seven departments were recruited. The process for examining the content validity of the instrument was presented clearly and in full.

The questionnaire survey, comprising the CoCB-NL, Work engagement scale, and Safety attitudes questionnaire were distributed electronically to a total population sample of 1730 employees, achieving a response rate of 63.2%, with 1062 complete cases. Survey 2 was completed by 590 respondents (48.4%).

Results for the different components of psychometric testing were presented clearly using Tables and narrative. The authors concluded that the CoCB-NL is sufficiently valid and reliable for assessing employees’ perceptions of their work environment and there was goodness of fit for a four- and five- factor model. Taking this forward to examine and improve workplace culture is imperative for the reasons outlined above and addressed in the manuscript.

Reviewer 2:

The authors provide a comprehensive demonstration of the validity and reliability of the CoCB-NL as an effective tool for assessing employers' perceptions of their work environment. While the methods employed are generally well-presented, further information is needed in order to replicate this study successfully. Specifically, the authors should clarify the sampling method used and enhance transparency regarding the research team responsible for content validation and its corresponding results.

Reviewers' comments:

Reviewer's Responses to Questions

**Comments to the Author**

1. Is the manuscript technically sound, and do the data support the conclusions?

Reviewer #1: Yes

Reviewer #2: Yes

2. Has the statistical analysis been performed appropriately and rigorously? 

Reviewer #1: Yes

Reviewer #2: Yes

3. Have the authors made all data underlying the findings in their manuscript fully available?

Reviewer #1: No

Reviewer #2: Yes

4. Is the manuscript presented in an intelligible fashion and written in standard English?

Reviewer #1: Yes

Reviewer #2: Yes

5. Review Comments to the Author

Reviewer #1: Thank you for inviting me to review this manuscript.

Recognising the centrality of a work environment that cares for and values its workforce for recruiting and retaining the workforce, achieving a high-performing organisation, and for safe and high-quality patient care, the aim of this study was to test the psychometric properties of a Dutch version of the Culture of Care Barometer.

The abstract was presented using headers and core details were included. There was no reference to a methodological framework for conducting this study.

The Introduction was succinct in contextualising this study- what characterises a positive work environment and why a positive work environment has positive outcomes for staff, patients, and wok organisations.

As a foundation for this study the authors conducted a systematic review on the psychometric evaluation of instruments measuring the work environment of healthcare professionals in hospitals, this identified the Culture of Care Barometer (Rafferty et al 2017) as an instrument that was worthy of further investigation to determine its validity and reliability for use in the Netherlands. Following on from this was a Delphi study by the authors that culminated in confirmation of 36 items by the experts.

For cross-cultural validation, the content and structural validity and internal consistency of the CoCB-NL were examined. The instrument was further validated by assessing convergent validity, discriminant validity, and responsiveness.

The design involved translation and cultural adaptation and a longitudinal survey to confirm structural validity and internal consistent, testing hypotheses to determine the convergent and discriminant validity and assessment of responsiveness.

The study was developed and reported according to the Consensus-Based Standards for the Selection of Health Measurement Instruments (COSMIN) study design checklist [19] and reporting guideline [22].

2 Dutch hospitals participated- the sampling approach was not reported- and a convenience sample of seven departments were recruited. The process for examining the content validity of the instrument was presented clearly and in full.

The questionnaire survey, comprising the CoCB-NL, Work engagement scale, and Safety attitudes questionnaire were distributed electronically to a total population sample of 1730 employees, achieving a response rate of 63.2%, with 1062 complete cases. Survey 2 was completed by 590 respondents (48.4%).

Results for the different components of psychometric testing were presented clearly using Tables and narrative. The authors concluded that the CoCB-NL is sufficiently valid and reliable for assessing employees’ perceptions of their work environment and there was goodness of fit for a four- and five- factor model. Taking this forward to examine and improve workplace culture is imperative for the reasons outlined above and addressed in the manuscript.

Reviewer #2: The authors provide a comprehensive demonstration of the validity and reliability of the CoCB-NL as an effective tool for assessing employers' perceptions of their work environment. While the methods employed are generally well-presented, further information is needed in order to replicate this study successfully. Specifically, the authors should clarify the sampling method used and enhance transparency regarding the research team responsible for content validation and its corresponding results

6. PLOS authors have the option to publish the peer review history of their article (what does this mean?). If published, this will include your full peer review and any attached files.

Reviewer #1: No

Reviewer #2: **Yes: **Helina Abebe Kurbi

---

## [Author Response · Author response to Decision Letter 0]

3 Sep 2023

Dear Prof. Ulutaş Deniz

We are pleased to learn that PLOS ONE is interested in publishing our paper ‘Measuring the work environment among health care professionals: validation of the Dutch version of the Culture of Care Barometer’. We thank both reviewers for providing us with the positive feedback and constructive suggestions to improve our manuscript. We have revised our manuscript accordingly. Below we provide a point-by-point reaction to the comments of each reviewer.

Journal Requirements

We have reviewed the manuscript for style compliance and adjusted the paper to adhere to the style requirements of PLOS ONE. We adjusted the supporting information from one file into three separate files with corresponding file names. 

In our Data Availability Statement we have selected the box ‘No - some restrictions will apply’. 

All data underlying reported findings are anonymized deposited with the Tilburg University data repository and access can be granted by either the authors or the data steward of Tilburg University (datasteward@tilburguniversity.edu), who will ask for permission of the Erasmus MC Board of Directors as they are the owners of the data. The accession number is currently pending, but we will of course provide it as soon as it is known. We expect to have that settled this month (September 2023).

Thank you for your check on the reference list. We have replaced the retracted reference (No. 37: Huebner LA, Zacher H. Effects of Action Planning After Employee Surveys. J Pers Psychol. 2022;21(1):23-36.) with the correct reference from the same authors (Huebner LA, Zacher H. The role of mean item ratings, topic distance, direct leadership, and voice climate in action planning after employee surveys. Acta Psychol (Amst). 2023;238:103950. PubMed PMID: 37379784). 

In addition, the reference of Guo et al. (2022; #27) has been updated from the 'EPub reference’ to the final reference of the paper. 

Both changes are not visible with track changes in the revised version because the whole reference list has been re-uploaded via Endnote.

Editor Comments:

Manuscript ID PONE-D-23-07783 entitled "Measuring the work environment among healthcare professionals: validation of the Dutch version of the Culture of Care Barometer" which you submitted to the PLOS ONE, has been reviewed. The comments of the reviewer(s) are included at the bottom of this letter.

The reviewer(s) have recommended publication, but also suggest some revisions to your manuscript. Therefore, I invite you to respond to the reviewer(s)' comments and revise your manuscript.

We thank the editor, Prof. Ulutaş Deniz, for the positive reaction. We will respond to the reviewers suggestions below and revise our manuscript as suggested by the reviewers. 

Reviewer 1:

Recognising the centrality of a work environment that cares for and values its workforce for recruiting and retaining the workforce, achieving a high-performing organisation, and for safe and high-quality patient care, the aim of this study was to test the psychometric properties of a Dutch version of the Culture of Care Barometer.

We thank reviewer 1 for this critical reflection on our objective. Indeed, this is exactly what we aim to do.

The abstract was presented using headers and core details were included. There was no reference to a methodological framework for conducting this study.

We have used a structured form to present our abstract, we thank reviewer 1 for pointing this out. 

Indeed, we do not refer in the abstract to a methodological framework for conducting this study. We used the COnsensus-based Standards for the selection of health Measurement Instruments (COSMIN) framework for measurement properties in this study and applied both the COSMIN study design checklist [1] and the reporting guideline [2]. PLOSONE’s reporting guidelines prescribe that references should not be included in the abstract. Therefore, we did not include the abbreviation COSMIN in the abstract. However, we do describe in the abstract the measurement properties that we have investigated with this study (content validity, structural validity, internal consistency, hypothesis testing for construct validity, and responsiveness). In the paper the COMSIN methodology and explanation of the various measurement properties are presented and explained in line 95-96, Figure 1 and lines 130-131, 141-142, 195-212. We trust that this explanation will be sufficient for the reviewer. 

The Introduction was succinct in contextualising this study- what characterises a positive work environment and why a positive work environment has positive outcomes for staff, patients, and wok organisations.

We thank reviewer 1 for this compliment.

As a foundation for this study the authors conducted a systematic review on the psychometric evaluation of instruments measuring the work environment of healthcare professionals in hospitals, this identified the Culture of Care Barometer (Rafferty et al 2017) as an instrument that was worthy of further investigation to determine its validity and reliability for use in the Netherlands. Following on from this was a Delphi study by the authors that culminated in confirmation of 36 items by the experts.

We thank reviewer 1 for this reflection on our approach.

For cross-cultural validation, the content and structural validity and internal consistency of the CoCB-NL were examined. The instrument was further validated by assessing convergent validity, discriminant validity, and responsiveness.

The design involved translation and cultural adaptation and a longitudinal survey to confirm structural validity and internal consistent, testing hypotheses to determine the convergent and discriminant validity and assessment of responsiveness.

 The study was developed and reported according to the Consensus-Based Standards for the Selection of Health Measurement Instruments (COSMIN) study design checklist [19] and reporting guideline [22].

We thank reviewer 1 for the reflection on our methodology used for the validation of the CoCB-NL. 

2 Dutch hospitals participated- the sampling approach was not reported- and a convenience sample of seven departments were recruited.

We acknowledge that the sampling approach for the two participating Dutch hospitals is not reported. We used the professional network of our research team members and purposively approached the hospitals that participated in our study as these hospitals were planning to improve their work environment. We have added this information in our manuscript (line 133-134).

The process for examining the content validity of the instrument was presented clearly and in full.

We are grateful for these compliments.

The questionnaire survey, comprising the CoCB-NL, Work engagement scale, and Safety attitudes questionnaire were distributed electronically to a total population sample of 1730 employees, achieving a response rate of 63.2%, with 1062 complete cases. Survey 2 was completed by 590 respondents (48.4%).

We thank reviewer 1 for the reflection on our survey.

Results for the different components of psychometric testing were presented clearly using Tables and narrative. The authors concluded that the CoCB-NL is sufficiently valid and reliable for assessing employees’ perceptions of their work environment and there was goodness of fit for a four- and five- factor model. Taking this forward to examine and improve workplace culture is imperative for the reasons outlined above and addressed in the manuscript.

We appreciate the acknowledgment of our efforts in clearly presenting the results of the psychometric testing. 

Reviewer 2:

The authors provide a comprehensive demonstration of the validity and reliability of the CoCB-NL as an effective tool for assessing employers' perceptions of their work environment. 

We thank reviewer 2 for the insightful feedback on our study. We are gratified that the reviewer recognizes the comprehensive nature of our demonstration of the validity and reliability of the CoCB-NL as a valuable instrument for evaluating employees' perceptions of their work environment.

While the methods employed are generally well-presented, further information is needed in order to replicate this study successfully. Specifically, the authors should clarify the sampling method used and enhance transparency regarding the research team responsible for content validation and its corresponding results.

We acknowledge the reviewer's concern regarding the replicability of our study and the necessity for additional information. Hence, we provided more detailed information of our sampling approach. The sample for translation and the content validation is clarified more clearly in line 142, 149-150. The supporting information ‘S1 Table’ includes descriptive data on the content validity sample. We have now referred to this more explicitly in line 152.

Furthermore, we understand the reviewer's call for increased transparency regarding the research team responsible for content validation and the corresponding results. Initially, we included the table with this information in a supplementary file. To enable readers to gain a more comprehensive understanding of the study's execution and results, we have replaced this table into the manuscript (line 161, table 1). 

Subsequently, we have changed the numbering of the tables.

We greatly appreciate the reviewer's valuable input, which helps us refine the presentation and accessibility of our research. 

We hope these revisions will make our manuscript suitable for publication in your journal.

Sincerely, on behalf of all co-authors,

---

## [Decision Letter · Decision Letter 1]

20 Dec 2023

PONE-D-23-07783R1Measuring the work environment among healthcare professionals: validation of the Dutch version of the Culture of Care BarometerPLOS ONE

Dear Dr. Maassen,

Thank you for submitting your manuscript to PLOS ONE. After careful consideration, we feel that it has merit but does not fully meet PLOS ONE’s publication criteria as it currently stands. Therefore, we invite you to submit a revised version of the manuscript that addresses the points raised during the review process.

Dear Authors,

The reviewers have provided their comments and recommendations. Based on their feedback, you are encouraged to revise the manuscript, addressing and responding to each comment specifically.

Best regards,

We look forward to receiving your revised manuscript.

Kind regards,

Elif Ulutaş Deniz

Academic Editor

PLOS ONE

Journal Requirements:

**Additional Editor Comments:**

Dear Susanne M. Maassen,

Manuscript PONE-D-23-07783R1 titled " Measuring the work environment among healthcare professionals: validation of the Dutch version of the Culture of Care Barometer’ which you submitted to PLOS ONE, has been reviewed. The comments of the reviewer(s) are included at the bottom of this letter.

The reviewers suggest some minor revisions to your manuscript. Therefore, I invite you to respond to the reviewer(s)' comments and revise your manuscript.

Best regards,

Elif Ulutaş Deniz

Reviewers' comments:

Reviewer's Responses to Questions

**Comments to the Author**

1. If the authors have adequately addressed your comments raised in a previous round of review and you feel that this manuscript is now acceptable for publication, you may indicate that here to bypass the “Comments to the Author” section, enter your conflict of interest statement in the “Confidential to Editor” section, and submit your "Accept" recommendation.

Reviewer #2: All comments have been addressed

Reviewer #3: All comments have been addressed

2. Is the manuscript technically sound, and do the data support the conclusions?

Reviewer #2: Yes

Reviewer #3: Yes

3. Has the statistical analysis been performed appropriately and rigorously? 

Reviewer #2: Yes

Reviewer #3: Yes

4. Have the authors made all data underlying the findings in their manuscript fully available?

Reviewer #2: Yes

Reviewer #3: Yes

5. Is the manuscript presented in an intelligible fashion and written in standard English?

Reviewer #2: Yes

Reviewer #3: Yes

6. Review Comments to the Author

Reviewer #2: Addressing my concerns and clarifying them strengthens the value of disseminating the findings. Consider knowledge translation programs for a broader impact.

Reviewer #3: Title

Measuring the work environment among healthcare professionals: validation of the Dutch version of the Culture of Care Barometer Capital Validation(V)

Abstract

WE

What are the inclusion and exclusion criteria?

Please state the full meaning. For example, in the first paragraph.

Example: A positive work environment (WE) is paramount for healthcare employees to provide good quality care.

Keyword

Too many keywords (10). Please identify the important keyword for this study.

Introduction

Need to explain first what Health literacy is.

Correct this. ( Sajjadi et al)

Results

0,30

For the tables, write the symbol. 0.30

Discussion

Three hypotheses were rejected because the CoCB-NL performed better than expected. Contrary to the expected moderate correlations, strong correlations were found between factors of the CoCB-NL and the SAQ-NL

Recommendation:

Make the discussion based on subheadings clearer to the reader.

Please harmonize this statement.

Recommend adding the subheading for recommendations for this study.

References

Most of the citations are above 5 years.

Very good.

7. PLOS authors have the option to publish the peer review history of their article (what does this mean?). If published, this will include your full peer review and any attached files.

Reviewer #2: **Yes: **Helina Abebe Kurbi

Reviewer #3: **Yes: **DR RUSNANI AB LATIF

---

## [Author Response · Author response to Decision Letter 1]

11 Jan 2024

Dear, Professor Ulutaş Deniz

We are pleased to learn that the revisions to our paper ‘Measuring the work environment among health care professionals: validation of the Dutch version of the Culture of Care Barometer’ has been well received by you and both reviewers. We thank you and both reviewers for providing us with previous and current constructive feedback which helped us to improve our paper. We have revised our manuscript accordingly. Below we provide a point-by-point reaction to the comments.

Journal Requirements:

We have reviewed the reference list, no alterations are needed anymore. In the previous revision we replaced the retracted article with another relevant paper. To address feedback below we have added a reference to the reference list: Brubakk et al., 2021. Ref no: 37. 

Editor Comments:

Manuscript PONE-D-23-07783R1 titled " Measuring the work environment among healthcare professionals: validation of the Dutch version of the Culture of Care Barometer’ which you submitted to PLOS ONE, has been reviewed. The comments of the reviewer(s) are included at the bottom of this letter.

The reviewers suggest some minor revisions to your manuscript. Therefore, I invite you to respond to the reviewer(s)' comments and revise your manuscript.

We thank the editor for the review and the invitation to address the minor revisions as suggested by the reviewers. We hope that our response is satisfactory. 

Reviewer 2

We thank you for the constructive feedback and are pleased that the revisions that we have made meet the expectations. 

Addressing my concerns and clarifying them strengthens the value of disseminating the findings. Consider knowledge translation programs for a broader impact.

Commensurate the reviewer we believe that the Culture of Care Barometer has much potential for practical application in the daily practice of healthcare organizations. We therefore already have planned knowledge translation activities to disseminate the findings and bridge the gap between research and practice, e.g. final report summaries, presentations at conferences, workshops, a web page and press releases.

Reviewer 3

We thank you for the positive and constructive feedback. We have addressed the points raised by the reviewer.

Title

Measuring the work environment among healthcare professionals: validation of the Dutch version of the Culture of Care Barometer Capital Validation(V)

Thank you for noticing. We have changed this.

Abstract:

WE: Please state the full meaning. For example, in the first paragraph. Example: A positive work environment (WE) is paramount for healthcare employees to provide good quality care.

What are the inclusion and exclusion criteria?

Thank you for noticing. We have added the abbreviation between brackets and information on the inclusion criteria for participating departments. 

Keyword

Too many keywords (10). Please identify the important keyword for this study.

We have identified five most reflective keywords for our study: ‘working conditions’, ‘organizational culture’, ‘hospitals’, ‘validation study’ and ‘questionnaire’. 

Introduction

Need to explain first what Health literacy is.

Correct this. ( Sajjadi et al)

We have not used the concept ‘health literacy’ nor the reference to Sajjadi et al in the introduction or study. Please, can the reviewer explain what is meant by this suggestion?

Results

0,30: For the tables, write the symbol. 0.30

Thank you for noticing. We have exchanged the comma’s for dots in table 1 and checked the other tables for inconsistencies. 

Discussion

Recommendation: Make the discussion based on subheadings clearer to the reader. 

Thank you for your suggestion, we now have added subheadings to enhance the readability. 

Discussion:

Three hypotheses were rejected because the CoCB-NL performed better than expected. Contrary to the expected moderate correlations, strong correlations were found between factors of the CoCB-NL and the SAQ-NL. Please harmonize this statement.

We agree with the reviewer that our formulation was not correct. We rephrased the statement into: ‘Three hypotheses were rejected because contrary to the expected moderate correlations, strong correlations were found between factors of the CoCB-NL and the SAQ-NL. This is commensurate findings of Brubakk et al. [37] who found that positive work environments associate with a positive safety climate and favorable patient safety outcomes.’(lines 334-338).

Recommend adding the subheading for recommendations for this study.

We have now added ‘recommendations’ to the conclusion subheading. 

References

Most of the citations are above 5 years. Very good.

Thank you for this remark. 

We have checked the figures with PACE.

We greatly appreciate the reviewer's valuable input, which helped us refine the presentation and accessibility of our research. 

We hope these revisions will make our paper suitable for publication in your journal.

Sincerely, on behalf of all co-authors,

Susanne Maassen

---

## [Decision Letter · Decision Letter 2]

24 Jan 2024

Measuring the work environment among healthcare professionals:

Validation of the Dutch version of the Culture of Care Barometer

PONE-D-23-07783R2

Dear Dr. Maassen,

We’re pleased to inform you that your manuscript has been judged scientifically suitable for publication and will be formally accepted for publication once it meets all outstanding technical requirements.

Kind regards,

Elif Ulutaş Deniz

Academic Editor

PLOS ONE

Additional Editor Comments (optional):

Reviewers' comments:

Reviewer's Responses to Questions

**Comments to the Author**

1. If the authors have adequately addressed your comments raised in a previous round of review and you feel that this manuscript is now acceptable for publication, you may indicate that here to bypass the “Comments to the Author” section, enter your conflict of interest statement in the “Confidential to Editor” section, and submit your "Accept" recommendation.

Reviewer #3: All comments have been addressed

2. Is the manuscript technically sound, and do the data support the conclusions?

Reviewer #3: Yes

3. Has the statistical analysis been performed appropriately and rigorously? 

Reviewer #3: Yes

4. Have the authors made all data underlying the findings in their manuscript fully available?

Reviewer #3: (No Response)

5. Is the manuscript presented in an intelligible fashion and written in standard English?

Reviewer #3: Yes

6. Review Comments to the Author

Reviewer #3: Dear Authors

Congratulations on your manuscript, I have checked the reviewer’s comment, and all the amendments were implemented in response to the reviewers' recommendations and commendations.

7. PLOS authors have the option to publish the peer review history of their article (what does this mean?). If published, this will include your full peer review and any attached files.

Reviewer #3: **Yes: **DR RUSNANI AB LATIF

---

## [Editor Report · Acceptance letter]

16 Feb 2024

PONE-D-23-07783R2 

PLOS ONE

Dear Dr. Maassen, 

I'm pleased to inform you that your manuscript has been deemed suitable for publication in PLOS ONE. Congratulations! Your manuscript is now being handed over to our production team.

Kind regards, 

on behalf of

Dr. Elif Ulutaş Deniz 

Academic Editor

PLOS ONE